# 3D CNN for neuropsychiatry: Predicting Autism with interpretable Deep Learning applied to minimally preprocessed structural MRI data

**Mélanie Garcia**[1,2], **Clare Kelly**[1,2,3]*

1 Department of Psychiatry at the School of Medicine, Trinity College Dublin, Dublin, Ireland, 2 Trinity College Institute of Neuroscience, Trinity College, Dublin, Ireland, 3 School of Psychology, Trinity College Dublin, Dublin, Ireland

* clare.kelly@tcd.ie

**Data Availability Statement:** All relevant data are within the manuscript and its Supporting Information files.

## Abstract

Predictive modeling approaches are enabling progress toward robust and reproducible brain-based markers of neuropsychiatric conditions by leveraging the power of multivariate analyses of large datasets. While deep learning (DL) offers another promising avenue to further advance progress, there are challenges related to implementation in 3D (best for MRI) and interpretability. Here, we address these challenges and describe an interpretable predictive pipeline for inferring Autism diagnosis using 3D DL applied to minimally processed structural MRI scans. We trained 3D DL models to predict Autism diagnosis using the openly available ABIDE I and II datasets (n = 1329, split into training, validation, and test sets). Importantly, we did not perform transformation to template space, to reduce bias and maximize sensitivity to structural alterations associated with Autism. Our models attained predictive accuracies equivalent to those of previous machine learning (ML) studies, while sidestepping the time- and resource-demanding requirement to first normalize data to a template. Our interpretation step, which identified brain regions that contributed most to accurate inference, revealed regional Autism-related alterations that were highly consistent with the literature, encompassing a left-lateralized network of regions supporting language processing. We have openly shared our code and models to enable further progress towards remaining challenges, such as the clinical heterogeneity of Autism and site effects, and to enable the extension of our method to other neuropsychiatric conditions.

## 1. Introduction

Autism Spectrum Disorder (Autism) is a complex and heterogeneous neurodevelopmental condition characterized by divergence from typical development on a number of behavioral dimensions, including communication, social interaction, and repetitive or restricted behaviors or areas of interest [1]. These behaviors likely reflect developmental neurological alterations over the lifespan [2–7], a suggestion supported by structural MRI (sMRI) studies [2–4,

**Funding:** MG received an Irish Research Council Postgraduate Scholarship (Project ID: GOIPG/2021/1508). We also received a donation for this research project from a French non-profit endowment fund called the HyperCube Institute (IHC). The IHC had to terminate all activity in 2020 due to the impacts of COVID-19. The funders had no role in study design, data collection and analysis, decision to publish, or preparation of the manuscript.

**Competing interests:** The authors have declared that no competing interests exist.

**Abbreviations:** CNN, Convolutional Neural Networks, a category of deep learning algorithm; ML, machine learning; DL, deep learning; Med3dNet, Resnet50: pretrained Residual Networks model with 50 layers; DenseNet121, Densely Connected Convolutional Networks with 121 layers; Epoch, a hyperparameter that defines the number of times that the learning algorithm has optimized the parameters on the entire training dataset; MRI, Magnetic Resonance Imaging.

8–24]. Despite substantial research effort, however, no compelling brain-based biomarkers have yet emerged. Instead, diagnosis of Autism Spectrum Disorder relies on clinician judgment and established observational tests like the Autism Diagnostic Observation Schedule (ADOS) [25] and the Autism Diagnostic Interview-Revised (ADI-R) [26]. Children are typically aged around 43 months when diagnosed [27], equivalent to over three and a half years, a situation reflecting the lengthy duration and the challenges posed by limited access to specialized clinicians. The significant heterogeneity inherent to Autism, coupled with the diverse range of long-term outcomes, underscores the need for reliable, reproducible brain biomarkers. Such biomarkers could refine diagnostic and treatment strategies, leading to improved outcomes. The development and application of predictive models could further assist clinicians in crafting personalized care plans for individuals with ASD [28].

A key challenge in the search for biomarkers and in the development of predictive models for Autism is the attainment of sample sizes that afford adequate statistical power. This challenge is exacerbated by the clinical heterogeneity of the condition [28]. Collaborative multisite studies, like ABIDE I and II [29, 30], have helped mitigate this challenge by providing well-powered samples. Analyses of these samples suggest a distributed pattern of Autism-related structural alterations [14–16, 18, 19, 22, 31, 32]. For instance, research on gyrification differences in Autism reveals distinct brain folding patterns in regions related to social cognition, communication, and sensory processing in Autistic individuals, compared to neurotypicals [16, 18–20, 23, 31–41]. These variations may influence the number, size, and depth of cortical folds. Atypical gyrification patterns can disrupt brain connectivity and communication, thereby modulating cognitive and sensory processing. The application of sophisticated multivariate approaches, such as Machine Learning (ML) and Deep Learning (DL), offer another promising avenue toward dissecting the complex neural signatures of Autism. ML encompasses a range of algorithms including DL and traditional multivariate methods; DL focuses on layered neural networks for complex pattern recognition, while traditional multivariate approaches rely on simpler, predefined statistical techniques for data analysis such as MANOVA (Multivariate Analysis of Variance).

These methods enable the simultaneous exploration of a very large set of features, offering much more powerful analytical capacity than univariate approaches. To date, such approaches have had moderate success, with recently reported prediction accuracies (for Autism diagnosis) in the range of 65–70% for models built using both functional and structural MRI data [42–45]. In an effort to boost accuracy through competition, Traut et al. [46] held an international challenge in which competing teams predicted Autism diagnosis using a large multisite dataset comprising preprocessed sMRI and rs-fMRI data from > 2,000 individuals. Of the 589 models submitted, the 10 best were combined and evaluated using a subset of unseen data (from one of the sites included in the main dataset), as well as data from an additional, independent acquisition site. This aggregated model achieved an ROC AUC of ~0.66, using features extracted from anatomical data only. Notably, this study highlighted two key insights: prediction accuracy improves with larger sample sizes, and while accuracy on unseen data matched validation accuracy, performance dropped on data from the new site, illustrating the challenge of generalizability, particularly to new data collection sites.

Despite recent progress in prediction accuracy, machine learning studies in this field face two significant limitations. The first is that preprocessing pipelines often involve many steps, each of which can introduce biases to prediction models. In particular, brain imaging studies typically rely on template registration methods using 'standard' brain models, such as the MNI152 template brain, constructed from demographically homogeneous non-clinical adult populations. By warping the brains of those with a clinical diagnosis to fit a 'normal' template, this step may obscure structural or functional differences unique to these populations. In

addition to negatively impacting our ability to detect Autism-related alterations in brain structure, normalization to a template may also introduce biases, and lead to poorer reproducibility [28]. A second limitation is that datasets used for prediction tend to be clinically heterogeneous, but this heterogeneity is not explicitly accounted for in the models, leading to inconsistent results across datasets [47]. Many Autistic participants have comorbid conditions such as ADHD, anxiety, epilepsy, or Fragile X syndrome [15, 22, 24]. Not accounting for these comorbidities might introduce biases or yield non-specific biomarkers [15], since the label "Autism" in such analyses is not precisely defined.

Here, we aimed to address these critical gaps in neuroimaging analysis for Autism and neurodivergent populations more broadly. Our work introduces an innovative method that moves away from the conventional *template registration* paradigm. While computational efficiency is a beneficial aspect of our method, particularly for large-scale neuroimaging datasets, our primary focus was to develop an approach that minimized the impact on data preprocessing steps on our ability to detect neurodivergence, to find patterns that could help develop more nuanced diagnostic tools, thus enriching our understanding of Autism. To do this, we trained 3-dimensional DL models to predict Autism diagnosis from minimally preprocessed structural MRI data in native space. To mitigate the impact of clinical heterogeneity, we developed our models using a large sample of 1329 patients (521 with Autism) without comorbid conditions, adhering to the classical training-validation-testing framework. To evaluate the robustness of our models in the presence of comorbidities, we evaluated the three best-performing models on a second dataset comprising 270 patients (155 with Autism) with comorbid diagnoses.

While 2D DL models are popular for medical imaging applications, 3D DL is not yet widely used, in part, due to the large number of parameters required to optimize the models (far greater than in 2D) and concerns related to interpretability. To overcome the interpretability challenge, we used recently developed methods to create an interpretation pipeline that identifies predictive brain regions while avoiding the requirement for template normalization.

In the proof-of-concept analyses described below, our models matched the prediction accuracies typically obtained using machine learning (ML) models, while sidestepping potential biases from template normalization. The interpretation pipeline reliably identified a consistent set of brain regions across datasets (including participants with comorbidities) and models, aligning with prior structural imaging research on Autism.

To support ongoing advancements in this field, we have shared all our code publicly on GitHub (https://github.com/garciaml/Autism-3D-CNN-brain-sMRI), fostering further research and development of our methodology.

## 2. Materials and methods

### 2.1 Data and quality control

We used T1-weighted structural MRI data from the ABIDE I (980 scans) and II (857 scans) datasets [29, 30] and 140 scans from ADHD200 [48]. Quality control was conducted using BrainQCNet [49]; scans with a probability score below 60% were retained, as advised in [49]. This process resulted in 797 scans from ABIDE I, 704 from ABIDE II and 98 from ADHD200 for our analysis.

Our primary analysis was restricted to (1) participants with a diagnosis of Autism with no reported comorbid conditions and (2) comparison participants with no psychiatric diagnosis. Following the exclusion of participants with comorbidities, a dataset comprising 1329 participants was used for training, validating and testing our models.

The testing set, comprising 65 participants (26 diagnosed with Autism), was exclusively drawn from independent data collection sites not represented in the training (1074

participants, 421 with Autism) and validation (190 participants, 74 with Autism) sets. This approach ensured a distinct separation between training/validation and testing datasets.

To assess the effect of comorbidities on our models' prediction accuracy, we constructed a secondary evaluation set. This set included participants diagnosed with Autism who also had comorbid diagnoses such as ADHD, phobias, depression, and anxiety. This second testing set comprised 270 participants, 155 of whom were diagnosed with Autism.

For comprehensive information on the datasets, including participant selection and data preparation, refer to the supplementary section, tables in S1 and S2 Tables.

## 2.2 Preprocessing

We employed a minimal preprocessing pipeline that intentionally avoided transforming brain images to a standard template space. This decision was made to minimize any potential distortion of Autism-related structural variations due to the normalization process. Our methodology began with the application of FSL's Brain Extraction Tool (BET; available at https://fsl.fmrib.ox.ac.uk/fsl/fslwiki/BET), which removes non-brain tissue from the images. Following this, we performed a series of non-deforming adjustments to the data to prepare it for analysis by our deep learning algorithm:

- *Resolution homogenization*: the ABIDE datasets comprise data from different data collection sites, each of which has different scanners and acquisition protocols. This diversity results in T1-weighted volumes with varying voxel spacings, a factor that could introduce bias in our analysis. To address this, we resampled all volumes to a fixed resolution of 1.5mm*1.5mm*1.5mm, using Linear Interpolation and the Resample function from the Python library TorchIO (https://torchio.readthedocs.io/_modules/torchio/transforms/preprocessing/spatial/resample.html#Resample), built from the Insight Toolkit (https://itk.org/Doxygen/html/index.html). We also reordered the data to RAS+ orientation.

- *Intensity normalization*: To mitigate the impact of noise arising from voxel value outliers in each image, we truncated intensity values so that they fell within the 0.5 to 99.5 percentile range, using the RescaleIntensity function from TorchIO. We also normalized each volume by z-scoring, i.e., by subtracting the mean intensity value $v_m$ to each voxel value $v_i$, and then dividing by the standard deviation $v_{sd}$, obtaining a new voxel value $v'_i$.

$$v'_i = (v_i - v_m)/v_{sd}$$

- *Cropping or Padding*: To ensure uniform size across all volumes, we implemented a cropping and padding procedure that standardized each volume to 256 x 256 x 256 voxels. This dimension was chosen for two reasons. First, it is large enough to encompass the entire brain, thereby preserving all the structural information in each scan. Second, this size is well-suited as an input for our deep learning (DL) models, considering the specifications of the filters used throughout each network layer (described in detail below). An example of a scan preprocessed using this pipeline is provided in S12 Fig.

## 2.3 Classification models in 3D

To identify the most effective algorithm for our task and to detect potential overfitting [50], we compared two models: (1) DenseNet121 [51] and (2) Med3D-ResNet50 [52]. Both models are based on established CNN architectures with robust 2D performance [51, 52].

- DenseNet121 [51]: This model is known for its compact structure. It has fewer parameters than ResNet50, which makes it more manageable for training on 3D data. DenseNet121's architecture is designed to facilitate efficient information flow between layers, a feature particularly beneficial for processing complex neuroimaging data.

- Med3D-ResNet50 [52]: This variant of ResNet50 has been pre-trained on a range of medical images, including brain sMRI scans. The pre-training aspect of Med3D-ResNet50 generally allows for better model convergence and performance when applied to new but contextually similar data and tasks. For our specific application, we fine-tuned the Med3D-ResNet50 model, focusing on the last convolutional layers (comprising the 4th convolutional block). Additionally, we appended a classifier block at the end, which includes a global average pooling layer followed by a fully connected layer, to tailor the model to our task.

This approach of comparing and refining these models enabled us to determine the most suitable algorithm for analyzing our dataset, ensuring both accuracy and efficiency in our predictive modeling. Further details on the architecture of these models can be found in S3 and S4 Tables.

As in Huang et al. [51] and Chen et al. [52], we used the ReLU function as the activation function, the cross-entropy loss, and the Adam optimizer with a fixed learning rate of 0.001.

Model performance was evaluated using ROC AUC and accuracy scores. Since predictions took the form of probability scores, we set a threshold of 0.5 for the "Autism diagnosis" class—across models, any prediction above that threshold was considered a positive prediction of Autism diagnosis. To assess the presence of a multi-site effect, ROC AUC and accuracy scores were computed separately for each data collection site.

## 2.4 Interpreting outcomes of Deep Learning algorithms

To address the interpretability challenge inherent to Deep Learning predictions from medical imaging data, we built an interpretation pipeline containing three key components, as described in the following sections.

**2.4.1 Guided Grad-CAM.**  To interpret and assess the effectiveness of our 3D deep learning (DL) models, we implemented Guided Grad-CAM [53], a method that integrates guided backpropagation [54] and Grad-CAM [53]. This technique provides an advantageous balance between the detailed insights from feature maps generated by interpretability algorithms and the efficiency of processing time. Guided Grad-CAM [53] mathematically combines the outcomes of both algorithms through an element-wise product, yielding a high-resolution map that not only captures fine-grained features but also differentiates between classes.

Here, for each CNN model trained (whether DenseNet121 or Med3DNet-ResNet50), we employed Guided Grad-CAM to create a unique "attention map" for each participant during the inference phase, specifically at the first layer of the CNN. These attention maps were tailored to match the resolution and voxel dimensions of the input scans. The voxel values within these maps represented "importance" scores, indicating the significance of each voxel in predicting Autism or non-Autism in the trained CNN model. This approach provided a nuanced

understanding of the model's decision-making process, highlighting the specific brain regions and features that the model deemed critical for accurate diagnosis.

**2.4.2 HighRes3Dnet.** As noted above, a key feature of our preprocessing pipeline was the deliberate omission of normalization to a group template. This approach, while preserving the unique structural characteristics of individual brains, presented a challenge in pinpointing the brain areas most predictive of diagnosis across different participants. We addressed this by segmenting individual scans into distinct anatomical units, and then integrating this segmentation with the mask $M$ generated in the previous step. For this segmentation, we employed HighRes3Dnet [55], a deep learning algorithm for segmenting brain MRI scans. This algorithm follows the Geodesic Information Flows (GIF) brain parcellation protocol (Version 3, available at http://niftyweb.cs.ucl.ac.uk/program.php?p=GIF; [56]). The GIF algorithm is designed to handle variations in brain morphology, making it highly suitable for studies involving atypical brain development, such as in Autism [56].

Each participant's brain scan was segmented using the HighRes3DNet algorithm. Prior to this segmentation, we standardized the scans to a voxel size of 1mm x 1mm x 1mm using linear interpolation. An example HighRes3DNet segmentation is provided in S13 Fig.

**2.4.3 Our interpretation methodology.** *Step 1*. First, we generated 3D attention maps for all participants using guided Grad-CAM. For each 3D attention map, we calculated a key threshold value, $q_{50\%}$, which is the median value of all the voxel intensities in the map. Using this median value, we created a binary mask, $M$. In this mask, all voxel values below $q_{50\%}$ were set to 0, marking them as less significant, while voxels with values greater than $q_{50\%}$ were set to 1, identifying them as areas of higher significance. This binary mask $M$ thus serves to distinguish between more and less relevant regions in the brain scans, based on the attention map generated by the model.

*Step 2*. Simultaneously, we deployed the HighRes3DNet algorithm on all participant scans that had undergone preprocessing as detailed in section 2.4.2. This step involved applying HighRes3DNet's advanced segmentation capabilities to each preprocessed scan, ensuring that the brain structures within each scan were accurately identified and delineated.

*Step 3*. The segmented images obtained from HighRes3DNet were then rescaled to a uniform size of 256 x 256 x 256, with each voxel measuring 1.5mm x 1.5mm x 1.5mm. This resampling was essential to align the resolution of these segmented images with that of the 3D attention maps and the corresponding 3D brain masks derived from the guided Grad-CAM algorithm.

The details of the transformations applied to each segmented image are encapsulated within the affine matrix of the transformed segmented image.

Mathematically, we note:

- $X = [x, y, z, 1]$, the column vector of the coordinates x, y, z of a voxel in a segmented image obtained with HighRes3DNet (voxel size: 1mm*1mm*1mm),

- $Y = [x', y', z', 1]$ the column vector of the coordinates x', y', z' of a voxel in the corresponding transformed segmented image (size: 256*256*256; voxel size: 1.5mm*1.5mm*1.5mm),

- $A \in |R^4$ the affine matrix of the transformed segmented image.

- $B$, the inverse matrix of $A$, such that $BA = A^{-1}A = I$, where $I$ is the identity matrix in $R^4$.

Thus, we have the relationship:

$$AX = Y$$

$$\Leftrightarrow X = BY, \forall (x', y', z') \in [1, 256]^3.$$

To establish a correlation between the attention maps from guided Grad-CAM and the segmented brain images, we used the voxel coordinates as a key link. For instance, if $x'$, $y'$, $z'$ are the coordinates of a voxel in a mask $M$ created by guided Grad-CAM, we can trace these coordinates back to their corresponding $x$, $y$, $z$ voxel locations in the segmented image. This allows us to retrieve both the voxel value and the specific brain area's name at the $(x, y, z)$ coordinates in the segmented image.

By applying this process to every scan, we obtained a count of voxels in each brain area delineated by HighRes3DNet that corresponded with the significant regions (where $M(x', y', z') = 1$) in a participant's brain mask.

This allowed us to build a table detailing the relative frequency for each brain area defined by the HighRes3DNet atlas. The relative frequency was calculated as the count of significant voxels $(x', y', z')$ in a given area where $M(x', y', z') = 1$, divided by the total voxel count in that area in the segmented image. This frequency essentially represents the proportion of each brain area that is deemed important for the prediction by the CNN model for a particular participant. This detailed breakdown allows for a nuanced understanding of how different brain areas contribute to the model's predictive process on an individual level.

*Step 4*. We used these calculated relative frequencies to perform a comparative analysis across different brain areas. This approach not only allowed us to examine the varying levels of significance attributed to each area by our models but also facilitated a hierarchical ranking of these brain regions. This ranking was established for each model and dataset, encompassing the training, validation, and testing phases.

We also extended this analysis to include different demographic categories, specifically examining gender (boys and girls) and various age groups (5–10 years, 10–15 years, 15–20 years, and over 20 years). This stratification allowed us to observe potential variations in brain area significance across these diverse segments.

Finally, we considered the type of prediction outcome–True Positives, True Negatives, False Positives, and False Negatives–to gain further insights into the model's performance. By scrutinizing the results across these various dimensions, our objective was to enhance the interpretability of our Convolutional Neural Network (CNN) models, providing a deeper understanding of how they process and evaluate neuroimaging data in the context of Autism diagnosis.

## 2.5 Machine and code availability

We trained our model on a GPU Nvidia RTX 3090 (24 GB memory) with a batch size of 2. We openly shared the code of this project on GitHub, in the repository: https://github.com/garciaml/Autism-3D-CNN-brain-sMRI. The models are also shared so that they can be reused as pre-trained models for similar applications.

## 2.6 Ethics statement

The databases used in the project—ABIDE 1 and 2, ADHD200—were made publicly available by the International Neuroimaging Data-sharing Initiative. All datasets were collected following local regulations on ethics and data protection. Each dataset had to be fully de-identified and anonymized, in accordance with the U. S. Health Insurance Portability and Accountability Act (HIPAA). Each research group tailored specific agreements on data reuse for their participants. Data usage is unrestricted for noncommercial research purposes, and is openly shared with the scientific community under the license Creative Commons BY-NC-SA. Our work with these open data was approved by the School of Psychology Research Ethics Committee at Trinity College Dublin.

## 3. Results

### 3.1 Training performance

Each model underwent training for up to 100 epochs. During this process, we assessed the accuracy of the models using the validation set, which consisted of 190 scans, at every two-epoch interval. Detailed information on the accuracy trends of the DenseNet161 and Med3d-ResNet50 models during the training phase can be found in S1 Fig.

For the ResNet50 model, the highest accuracy achieved on the validation set was 62.6%, recorded at 42 epochs. In the case of DenseNet121, the model reached an accuracy of 66.3% at 32 epochs and further improved to 67.4% at 70 epochs. Following this training phase, we proceeded to compare the performance of these three optimal models (one from ResNet50 and two iterations of DenseNet121) in terms of their predictive accuracy across the training, validation, and testing sets. This comparative analysis aimed to identify the most effective model configuration for accurately diagnosing Autism across different data sets.

### 3.2 Prediction performance: Autism diagnosis

In the task of predicting Autism diagnosis, the top three models exhibited distinct performance characteristics, as evidenced by their Receiver Operating Characteristic (ROC) curves presented in Fig 1. At 42 epochs, the Med3d-ResNet50 model showed signs of overfitting: it achieved high accuracy (94.2%) and ROC AUC (99.9%) on the training set but performed considerably lower on the validation (62.6% accuracy and 62.1% AUC) and testing sets (53.8% accuracy and 57.3% AUC). At 32 epochs, the DenseNet121 model demonstrated more consistent performance across the training (65.5% accuracy and 69.1% AUC), validation (66.3% accuracy and 68.8% AUC), and testing (55.4% accuracy and 60.7% AUC) sets. At 70 epochs, the DenseNet121 model exhibited a decline in performance on the testing set (40% accuracy and 38.1% AUC) despite showing improved performance on the training (69.7% accuracy and 77.1% AUC) and validation (67.4% accuracy and 68.1% AUC) sets compared to its 32-epoch counterpart.

Table 1 details the sensitivity and specificity of each model across different datasets. The DenseNet121-32ep model showed high specificity on the training and validation sets but lower sensitivity. Interestingly, it displayed high sensitivity but lower specificity on the testing set. DenseNet121-70ep presented a similar pattern on the testing set, whereas it maintained balanced and relatively high sensitivity and specificity on the training and validation sets. Med3d-ResNet50-42ep exhibited very high sensitivity and specificity on the training set, but

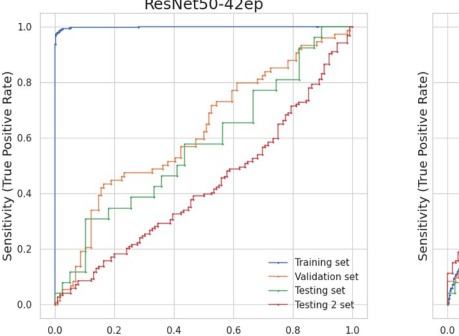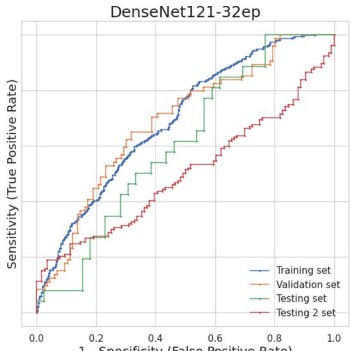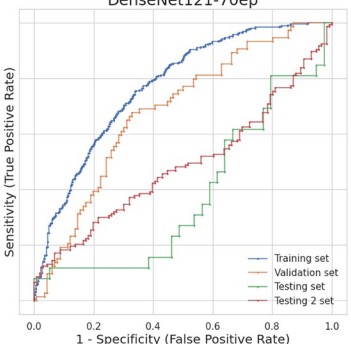

**Fig 1. Receiver operating characteristic curves for the three models across four datasets.**

**Table 1. Sensitivity and specificity of each model across datasets.** Datasets are Training, Validation, Testing (No Comorbidity), and Testing Set 2 (With Comorbidities).

| | Dataset | Med3dNet-Resnet50 on 42 epochs | DenseNet121 on 32 epochs | DenseNet121 on 70 epochs |
|---|---|---|---|---|
| **Sensitivity** | Train | **85,3%** | 32,8% | 68,2% |
| | Validation | 17,6% | 36,5% | **66,2%** |
| | Test | 50% | **84,6%** | 69,2% |
| | Test 2 | 8,4% | 7,6% | **31%** |
| **Specificity** | Train | **100%** | 86,7% | 70,8% |
| | Validation | **91,4%** | 85,3% | 68,1% |
| | Test | **56,4%** | 35,9% | 20,5% |
| | Test 2 | 87,8% | **100%** | 73% |

on the validation set, the model showed low sensitivity and very high specificity. On the testing set, the sensitivity and specificity were moderate but balanced.

Sensitivity was low for all models on the second testing set, which included participants with comorbidities. This highlights a significant challenge: models trained and tested on data from individuals without comorbidities struggle to accurately predict Autism diagnosis in individuals with comorbidities, leading to a substantial increase in False Negatives. This could be due to the diminished prominence of Autism-specific neuroimaging markers in the presence of comorbid conditions or the need for more diverse training data to effectively capture the full spectrum of Autism in the context of comorbidities.

For a comprehensive review of the models' performance, please refer to figures in S1–S4 Figs, and table in S5 Table. Additionally, figures in S5–S8 Figs. offer a comparison of the predicted scores with actual diagnostic scores, providing further insights into the models' effectiveness.

### 3.3 Interpretability: *True Positive* discriminative ROIs

We applied HighRes3DNet for segmentation (using GIF parcellation) on each participant's scan, aiming to quantify the "prediction importance" as determined by the guided Grad-CAM algorithm for our top three models. This process helped us identify the brain regions most influential in correctly predicting True Positives (TP), True Negatives (TN), False Positives (FP), and False Negatives (FN) across the combined dataset (training, validation, testing 1 & 2 sets).

For each model-dataset pair, we designated the "most predictive" regions as those whose relative frequency values exceeded the 90th percentile (refer to Section 3.6). This criterion identified 16 key regions per model-dataset pair. To draw comparisons across models and datasets (including training, validation, testing sets with and without comorbidities), we tallied the occurrence (denoted as 1) or absence (denoted as 0) of these most predictive regions, specifically focusing on True Positives and True Negatives. Across all models, 79 areas emerged as highly predictive for True Positives, comprising 26 bilateral regions, 23 exclusive to the left hemisphere, 3 unique to the right hemisphere, and the Corpus Callosum. When focusing on areas consistently replicated across all datasets, left hemisphere regions proved more replicable, with a significant concentration in the prefrontal cortex. Table in S6 Table summarizes the most consistent regions across models and datasets crucial for predicting True Positives, accompanied by an in-depth analysis.

In summary, 17 regions were consistently predictive of True Positives across all models and datasets. These are illustrated in Fig 2A and encompass areas in the left frontal lobe (including

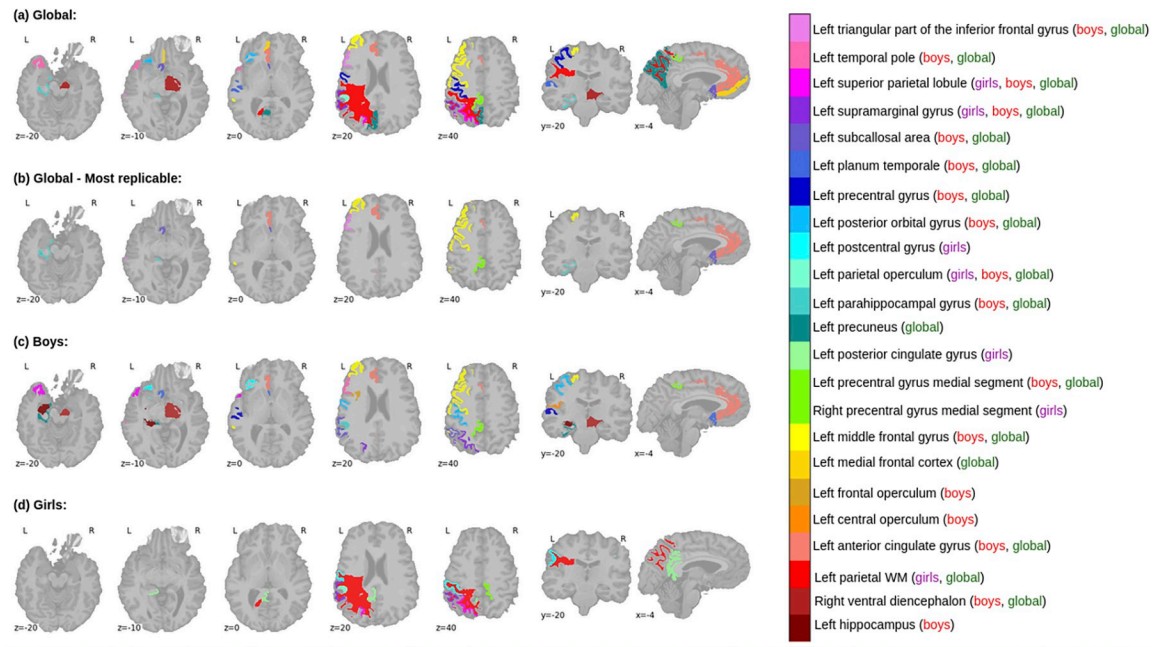

**Fig 2.** True Positive discriminative ROIs: (a) Regions most predictive of Autism diagnosis; (b) Regions most predictive across datasets; (c) Regions most predictive for boys; (d) Regions most predictive for girls.

medial frontal cortex, inferior and middle frontal gyrus, lateral and medial precentral gyrus, anterior and subcallosal cingulate gyrus, posterior orbital gyrus), left temporal lobe (temporal pole, planum temporale, parahippocampal gyrus), parietal lobe (parietal operculum, supra-marginal gyrus, superior parietal lobule), left parietal white matter, and right ventral thalamus.

Additionally, when analyzing the regions most predictive across datasets and consistently replicated across the three models, we predominantly found left hemisphere regions located in the frontal lobe, including the middle and inferior frontal gyrus (pars triangularis) and medial precentral gyrus, as well as in the limbic system and its associated structures, such as the anterior cingulate gyrus, subgenual cingulate gyrus, and parahippocampal gyrus (illustrated in Fig 2B).

**3.3.1 Effect of gender.** The brain regions important for predicting True Positives in Autism diagnosis showed notable differences between boys and girls. Shared regions for both genders were identified primarily in the left parietal lobe, including the parietal operculum, supramarginal gyrus, and superior parietal lobule (as illustrated in Fig 2C and 2D). Overall, the regions significant for predicting True Positives in boys exhibited higher consistency across the datasets (training, validation, testing 1/2), compared to those for girls.

For boys, a number of left prefrontal regions consistently emerged as predictive of Autism diagnosis. These included the left anterior cingulate gyrus, middle frontal gyrus, and the medial precentral gyrus. Specifically, the inferior frontal gyrus (pars triangularis) was identi-fied in the ResNet50 model at 42 epochs, while the medial precentral gyrus was highlighted in the DenseNet121 model at 32 epochs. Additionally, the precentral and parahippocampal gyrus were recognized in the DenseNet121 model at 70 epochs.

The gender-specific regions important for accurate Autism diagnosis are comprehensively presented in S10 and S11 Tables.

**3.3.2 Relationship with age.** Autism is associated with atypical brain development throughout an individual's life. To explore developmental patterns in brain regions most

predictive of Autism (True Positives), we divided our participants into four age groups (5–10 years, 10–15 years, 15–20 years, and over 20 years) and identified the key predictive regions within each group, separately for boys and girls. These findings are detailed in S12–S19 Tables.

This analysis revealed age-dependent variations in the most discriminative brain regions. For boys aged 5–10 years, the left precentral gyrus, central operculum, and posterior orbital gyrus were consistently predictive of True Positives. In contrast, for boys aged 10–15 years, the left inferior frontal gyrus (pars triangularis), subcallosal/subgenual cingulate cortex, and supra-marginal gyrus emerged as the most predictive regions.

Additionally, we noted a decline in the replicability of specific regions with increasing age. For instance, in boys aged 15–20 years, the most predictive regions included the left inferior frontal gyrus (pars triangularis), posterior orbital gyrus, and putamen, but this was predominantly in participants without comorbidities. In males aged over 20 years, the left temporal regions (parahippocampal gyrus, superior temporal gyrus, and temporal pole) were found to be most predictive, again in those without comorbid conditions.

A broader examination of prediction performance across these age groups revealed further trends. Notably, there was a decrease in the number of False Negatives and True Negatives with advancing age for both boys and girls. This pattern suggests an increase in sensitivity but a reduction in specificity of Autism diagnosis predictions with age. Such trends underscore the dynamic nature of brain changes in Autism and highlight the importance of age-specific approaches in understanding and predicting the condition.

**3.3.3 True Negatives.**   Using the methodology outlined earlier, we identified the brain regions that were most predictive of True Negatives (indicating the absence of an Autism diagnosis). The findings, detailed in S7 Table, revealed that the most consistently identified regions for predicting True Negatives were predominantly in the left hemisphere. These included the frontal operculum, precuneus, planum polare, inferior occipital gyrus, occipital fusiform gyrus, superior occipital gyrus, and the thalamus proper. Additionally, cerebellar vermal lobules VI and VII were also significant in this prediction.

Notably, we observed that some regions, such as the left precuneus, parietal operculum, superior parietal lobe, and the right thalamus, were critical (with varying degrees of replicability and across different models) for predicting both True Negatives and True Positives. This suggests that these areas play a complex role in the neural underpinnings of Autism.

The 23 other regions identified as crucial for predicting True Negatives differed from those significant for predicting True Positives. This finding aligns with expectations, indicating that the models tend to focus on distinct brain patterns when differentiating between class 0 (non-Autism) and class 1 (Autism). This highlights the nuanced perspective offered by our models for discerning varying neural signatures associated with Autism.

**3.3.4 Poor predictions—False Positives and False Negatives.**   We extended the previously described methodology to identify brain regions most predictive of False Positives (incorrect Autism diagnosis) and False Negatives (failure to correctly predict Autism diagnosis). The detailed findings are presented in S8 and S9 Tables. A notable observation was the absence of highly replicable False Positive regions across all datasets. This absence is intriguing, since it suggests a lack of distinct anatomical brain patterns consistently associated with False Positive predictions in our classification task. This finding indirectly reinforces the significance of the anatomical brain patterns identified as crucial for predicting True Positives.

For the DenseNet121-70 model, regions with a high level of replicability for False Positives showed some overlap with regions important for predicting True Positives in the other models. These regions included the middle frontal gyrus, medial segment of the precentral gyrus, and the triangular part of the inferior frontal gyrus. This overlap not only underscores the

differences in how each algorithm is calibrated but also highlights the value of comparing various models to gain a comprehensive understanding of their predictive behaviors.

Regarding False Negatives, the most replicable regions were predominantly located in the left hemisphere. These included the left frontal operculum, left precuneus, left superior temporal gyrus, left planum polare, left inferior occipital gyrus, and left occipital fusiform gyrus. The consistency of these regions in relation to False Negatives offers insights into potential neural areas that might be underrepresented or misinterpreted in the predictive models, particularly in cases where an Autism diagnosis is missed.

## 3.4 Does image background contribute to model predictions?

We investigated whether elements outside the brain, specifically the image background, influenced the models' predictions. We focused on the relative frequency (RF) of the background area in the models' predictive analyses.

For the Med3d-ResNet50-42ep model, the background had the lowest RF at 0.97%, indicating a minimal role in predictions. Similarly, for the DenseNet121-70ep model, the background's RF was 0.28%, the second lowest among the considered areas, suggesting it was not a significant predictive factor. With the DenseNet121-32ep model, the RF for background voxels was 0.74%, placing it among the least informative 4% of areas.

These findings reinforce the conclusion that our models predominantly relied on internal brain structures rather than external or background information in making predictions. This underscores their effectiveness and validity in focusing on relevant neuroanatomical features for Autism diagnosis.

## 3.5 Multi-site effect

We noted varying levels of consistency in the probability score distributions across different data collection sites, a phenomenon we refer to as the "multi-site effect" (detailed in S9–S11 Figs). To further examine this, we tabulated the accuracy scores for each site across the entire dataset, encompassing training, validation, and testing sets. These data are shown in S20 Table. The variation in accuracy scores across these sites not only highlights the multi-site effect but also emphasizes the need to consider site-specific factors in future analyses and model development.

## 4. Discussion

This study outlines and demonstrates a novel approach for inferring Autism diagnosis from structural brain imaging data using 3D deep learning (DL) algorithms. To enhance the interpretability of the model outputs, we also used a second type of algorithm—guided Grad-CAM [53]—to identify brain features contributing to the predictions This approach identified a cluster of brain regions primarily in the left hemisphere, including areas in the lateral and medial prefrontal cortex, anterior cingulate, superior temporal gyrus, and lateral parietal regions such as the supramarginal gyrus and parahippocampal gyrus. Notably, the right thalamus was the sole right hemisphere region emphasized in our analysis. The brain regions pinpointed through this interpretability analysis, which play crucial roles in the accurate prediction of Autism diagnoses (True Positives), align closely with existing neuroscientific literature. The success our predictive modeling framework demonstrates its significant potential for application to further datasets to identify and refine sensitive and specific brain biomarkers of Autism.

## 4.1 3D Deep Learning applied to minimally processed data

To our knowledge, this is the first time that 3D-DL CNNs have been used to predict Autism diagnosis from 3D structural MRI scans. Our results demonstrate that these algorithms can infer Autism diagnoses from structural MRI data with a level of accuracy comparable to, or even surpassing, traditional machine learning (ML) methods, while requiring fewer training epochs. The average accuracy score (64.1%) and ROC AUC score (0.67) obtained for participants without comorbidities is consistent with previous ML models trained on sMRI data (e.g., [46]). The significance of our findings is underscored when considering the efficiency of DL models compared to ML approaches. ML algorithms typically necessitate extensive preprocessing of MRI data, including normalization to a template space. In contrast, our DL models utilized minimally preprocessed data, notably circumventing the common requirement of transformation to template space —a step that could obscure detection of structural alterations relevant to the diagnosis.

Our preprocessing did include some essential steps to account for the variability in scanners and acquisition protocols across different data collection sites, which resulted in heterogeneous voxel spacing and signal intensities. We implemented resolution homogenization and intensity normalization to address these discrepancies. While these steps were necessary, they could potentially introduce bias into the algorithm. Additionally, we observed a distinct effect of the data collection site on our results. Future research in this area should consider incorporating advanced preprocessing techniques like the ComBat algorithm [57] to integrate scan parameters into the training process, aiming to minimize the impact of site-specific variations. Such improvements could further refine the accuracy and applicability of DL models in neuroimaging analyses for Autism and potentially other neurodevelopmental disorders.

## 4.2 Interpretability

The outputs of DL models are not straightforwardly understandable, giving rise to the challenge of poor interpretability. This difficulty stems from the mathematical structure of DL models, which consist of multiple nonlinear functions, each being a sum of other functions. In addition, models like the 3D CNNs used in our study require optimization of a large number of parameters. Addressing this challenge, our study aimed to develop a pipeline that enhances interpretability by extracting and analyzing predictive brain regions. We selected Guided Grad-CAM [53] for its efficiency in computation time and its ability to generate detailed class-specific segmentations of relevant voxels in the input images.

One of the key hurdles we faced was identifying predictive brain areas for Autism diagnosis across participants while bypassing the usual requirement for template normalization. To overcome this, we employed HighRes3Dnet [55], a segmentation algorithm known for its pathology-agnostic nature, robustness to brain morphology variations, and computational efficiency. This approach allowed us to conduct a thorough analysis of the most relevant regions for predicting Autism, examining true and false outcomes separately across datasets and algorithms. We also pinpointed regions that were consistently identified by different models and datasets. This detailed analysis is important because each model has biases, likely resulting in a differential weighting of anatomical features and brain areas.

Our analysis revealed that morphological features of certain regions, particularly in the left prefrontal cortex (including areas like the inferior and middle frontal gyrus, medial prefrontal gyrus, anterior and subgenual cingulate cortex), and the parahippocampal gyrus, contributed most to accurate predictions of Autism diagnosis. These findings align with existing literature on Autism-related disruptions in these areas in cortical development [22, 33–35] and gyrification processes [22, 36]. Furthermore, we observed variations in the most predictive regions in accordance with gender, age, and presence of comorbidities. This is consistent with

observations that Autism is a complex condition, with patterns of neurological divergence that vary with age [15, 22, 34, 35] and gender [15, 22, 58]. Notably, left parietal white matter emerged as crucial for accurate prediction of diagnosis in boys, implicating altered connectivity between visual and frontoparietal networks, which has been observed in sibling studies using functional connectomics approaches [59]. This supports the potential of structural MRI (sMRI) to complement functional MRI (fMRI) findings, emphasizing the need for multimodal studies in future Autism research.

Many of the left-hemisphere regions identified as contributing to accurate inference of Autism diagnosis fall within the canonical left-lateralized language network, including inferior prefrontal and inferior parietal regions, and the planum temporale in superior temporal gyrus [60–63]. Divergent structure and function in the language network is a robust and reproducible finding in Autism [38, 39, 61–64]. Since early language processing appears to be an important predictor of long-term outcomes in Autism [65–67] identification of early-emerging structural alterations in the underlying language network has the potential to yield a powerful marker of Autism or Autism subtypes, which could, in turn, direct individualized interventions and improve prognosis. Finally, our findings are also consistent with evidence of atypical socio-emotional and motor circuitry in Autism, involving areas such as the limbic system and dorsal medial frontal cortex [37, 68–71].

While our novel interpretation step successfully identified regions exhibiting morphological features that were relevant to the model-based inference of Autism, it could not provide information on what these morphological features were. Features such as cortical thickness, the location of the gray-white boundary, surface area, and gyral/sulcal morphometry could all play a role in prediction of Autism [20, 34, 40]; and different morphological features may be relevant in different brain areas. While the precise nature of the Autism-related morphological features are not discernable from our analyses, future research can extend our predictive modeling analyses with in-depth, targeted, and hypothesis-driven examinations of these areas in independent samples to uncover the nature of these features.

## 4.3 Limitations and ethics

Our pipeline for predicting neuropsychiatric diagnoses, particularly Autism, using minimally preprocessed T1 MRI scans represents a significant advancement in the application of interpretable 3D deep learning (DL) in biological psychiatry. It also contributes towards identifying consistent brain biomarkers, potentially refining diagnosis and treatment strategies across various conditions. However, there are several limitations to our study that offer opportunities for refinement in future research.

One limitation is that our models were trained over 100 epochs, a standard duration in studies using 3D MRI scans [72], but possibly insufficient for full convergence and optimization of the algorithms. Future studies could explore training over more extended periods or employ techniques like early stopping [73] to enhance model training efficiency. Additionally, using entire structural MRI scans for whole-brain prediction might have presented a challenge in converging towards an optimal solution.

Another critical limitation is the size and diversity of our dataset. While substantial (1,074 participants for training, 525 for validation and testing), it may not fully capture the clinical heterogeneity intrinsic to Autism. This limitation was evident in the relatively poor performance observed in our second test set, which included participants with comorbid diagnoses, yielding an average accuracy of 46.3%, ROC AUC of 0.47, and an average sensitivity of 15.7%. The challenge is compounded by ongoing debates in the field regarding the utility and appropriateness of a binary "Autism vs. non-Autism" classification, given the spectrum nature of

Autism, which may include multiple subtypes [21] and the overlap of symptoms and neuro-markers with other psychological conditions [15]. To address these issues, future research will require even larger and more diverse datasets. This expansion would enable a more nuanced examination of the clinical heterogeneity of Autism and potentially allow for the prediction of a broader range of categories beyond the binary Autism and non-Autism classification.

Another limitation is related to the segmentation algorithm we used in the interpretation step. We used HighRes3Dnet [55] to obtain rapid segmentation for each brain using the GIF algorithm [56], designed for robustness in atypically developing brains. However, the segmentation generated by this algorithm is relatively coarse, with large parcels that may encompass anatomically diverse regions like the anterior cingulate gyrus or superior parietal lobule. Additionally, while our interpretation process identified key regions for Autism prediction, it did not elucidate the specific predictive morphological features within these regions.

Performing classification in native space, as opposed to template space, is a unique aspect of our study. This approach can enhance the visibility of information sensitive to individual anatomical variability but may also introduce irrelevant variability to the classification task. For example, spatially normalized data in template space might lose detail or become distorted in native space, whereas unique anatomical features could be more pronounced. Further research is required to fully understand these effects.

Our study observed a multi-site effect in ASD prediction, aligning with findings from previous studies that registered brains to a common template. This effect is likely influenced by various factors, such as differences in image acquisition protocols, differences in the severity of ASD across sites, and disparities in sample sizes and diagnosis representation (e.g., there were 103 scans from NYU but only 5 from Stanford). Other factors include the presence and frequency of comorbidities, gender and age distribution, and potentially cultural differences across countries [59], though the exact impact of these is less clear. We did not quantify the influence of these factors in our study, mainly due to challenges like uneven sample sizes and limited meta-data. While equalizing sample sizes across sites could address some issues, it would also reduce the overall dataset size, limiting the breadth of our findings. Future research addressing these complexities will be crucial in advancing our understanding of ASD and refining DL-based predictive models in neuroimaging.

In navigating the ethical landscape of our research, particularly in the development of models for Autism spectrum disorder (ASD) diagnosis, it's crucial to consider the impact of our findings. This is especially important in situations where models demonstrate high sensitivity but low specificity. While such models are effective in detecting true positives (actual cases of ASD), they also increase the risk of false positives. In the context of Autism, false positives may carry very significant implications, leading to unwarranted labeling, unnecessary interventions or treatments, and the social stigma of a misdiagnosis. These potential consequences necessitate careful consideration and responsible handling.

On the other hand, the implications of false negatives—cases where Autism is present but goes undiagnosed—are also significant. Missing a true condition can lead to delays or absence of early intervention and support, which are crucial in improving outcomes in Autism. The ethical considerations of such outcomes are intricate and can significantly vary among individuals with Autism and in different contexts. These complexities underscore the need for a comprehensive ethical framework underpinning the creation and implementation of an image-based diagnostic tool for ASD. This framework should thoroughly evaluate and balance these trade-offs, prioritizing the minimization of harm and considering the broad spectrum and individual variability within ASD. Ensuring ethical integrity in the application of such diagnostic models is as important as their technical accuracy and effectiveness.

### 4.4 Future directions

There is considerable scope to extend our interpretable DL pipeline to the prediction of other neurological or neuropsychiatric conditions or to other MRI modalities. Traut et al. [46] reported that prediction of Autism was considerably improved (from AUC = 0.66 using only anatomical MRI to AUC = 0.79 using both anatomical and functional data) for a blended model that incorporated both functional and structural MRI data. Future studies will explore the potential benefits of incorporating functional MRI data into our models. We also plan to enhance our models by increasing the number of training epochs, experimenting with different architectures, integrating scanning parameters, and accounting for confounders like gender and age. Moreover, diversifying and expanding class labels could refine our approach. To facilitate collaborative advancement in this field, we have made our code available at https://github.com/garciaml/Autism-3D-CNN-brain-sMRI.

Another critical area of future research involves systematically comparing analyses in native space analysis with analyses in template space. This comparison, assessing metrics like classification accuracy, generalizability, and feature interpretability, will help quantify the advantages and drawbacks of each approach. Additionally, we will consider the computational demands and the necessity for more sophisticated data augmentation strategies in native space analyses. Such comprehensive evaluations are expected to yield guidelines for determining the optimal use of native space analysis in neuroimaging. Involving larger and more balanced datasets is crucial to fully understand and mitigate the sources of site effects in ASD imaging studies too.

Finally, our study, which focused on direct classification of MRI data for neuropsychiatric conditions like ASD, can be complemented by normative modeling. This approach involves creating models of typical brain development or structure and identifying individual deviations from these norms as potential indicators of pathology. Normative models, particularly useful for conditions that exhibit significant variability, like Autism, can be enhanced using unsupervised DL algorithms like autoencoders or Generative Adversarial Networks for dimensionality reduction. These methods can aid in building more nuanced, normative models that account for the wide spectrum of neurodevelopmental variations.

## 5. Conclusion

In this paper, we introduced an innovative approach to develop a predictive model for Autism diagnosis using 3D deep learning (DL) techniques applied to structural MRI scans. This was complemented by an interpretative method that effectively pinpointed the key brain regions crucial for accurate diagnosis. A distinctive aspect of our approach was the application of these models to minimally preprocessed data, bypassing the commonly used step of template normalization, which can potentially mask structural changes relevant to the diagnosis. Our findings show that the predictive efficacy of our models is on par with existing machine learning (ML) models, while also offering the advantage of faster prediction generation due to minimal preprocessing requirements. This opens up substantial opportunities for refining our methodology and integrating additional modalities, such as functional MRI (fMRI), to enhance predictive accuracy.

The interpretative aspect of our DL models identified brain regions with high predictive value that align with existing neuroscientific literature. This underscores the capability of 3D DL models to yield biologically plausible results without relying on pre-computed morphological derivatives like volumes, cortical thickness, or surface area, and without prior assumptions. While the clinical heterogeneity of Autism presents ongoing challenges, our decision to openly share our code and models paves the way for further research and development in this area and helps advance the field toward reliable and reproducible brain biomarkers for neuropsychiatric conditions.

## Supporting information

**S1 Table. Partitioning of the ABIDE I, ABIDE II, and ADHD200 datasets into training, validation and testing sets.**
(CSV)

**S2 Table. Gender breakdown and distribution of age and FIQ score for each dataset (training, validation, testing, testing 2 sets).**
(CSV)

**S3 Table. Configuration of DenseNet121 Model for 3D Scan Classification Task—Input Size: 256 x 256 x 256.**
(CSV)

**S4 Table. Architecture of ResNet50 for Autism prediction: Parameter extraction and fine-tuning details.** The layers for which we extracted the parameters from the pre-trained model Med3d are named "Fixed", and the layers for which we continued training the parameters to fine-tune the model and adapt it to the task of predicting Autism are named "Fine-tuned".
(CSV)

**S5 Table. Comparative performance of Autism prediction: Med3d-ResNet50 (42 epochs) vs. DenseNet121 (32 epochs) vs. DenseNet121 (70 epochs).**
(CSV)

**S6 Table. Best regions for predicting True Positives (TP, i.e. true diagnosis of Autism).** Each row is for one region, each column is for one model (R42 for ResNet50 trained on 42 epochs, D32 for DenseNet121 trained on 32 epochs, D70 for DenseNet121 trained on 70 epochs) and one combination of datasets considered (training+validation+testing 1 sets ("no comorb" for no comorbidity), or all these sets + testing set 2 ("with comorb" for containing subjects with comorbidities), each case returns the number of datasets where the region was important for predicting TP for the model considered.
(CSV)

**S7 Table. Best regions for predicting True Negatives (TN, i.e. no diagnosis of Autism).** Each row is for one region, each column is for one model and one combination of datasets considered (training+validation+testing 1 sets (no comorbidity), or all these sets + testing set 2 (containing subjects with comorbidities)), each case returns the number of datasets where the region was important for predicting TN for the model considered.
(CSV)

**S8 Table. Best regions for predicting False Positives (FP, i.e. prediction of Autism whereas no diagnosis Autism).** Each row is for one region, each column is for one model and one combination of datasets considered (training+validation+testing 1 sets (no comorbidity), or all these sets + testing set 2 (containing subjects with comorbidities)), each case returns the number of datasets where the region was important for predicting TN for the model considered.
(CSV)

**S9 Table. Best regions for predicting False Negatives (FN, i.e. no prediction of Autism whereas diagnosed Autism).** Each row is for one region, each column is for one model and one combination of datasets considered (training+validation+testing 1 sets (no comorbidity), or all these sets + testing set 2 (containing subjects with comorbidities)), each case returns the number of datasets where the region was important for predicting TN for the model

considered.
(CSV)

**S10 Table. Best regions for predicting True Positives (TP, i.e. true diagnosis of Autism) for boys.**
(CSV)

**S11 Table. Best regions for predicting True Positives (TP, i.e. true diagnosis of Autism) for girls.**
(CSV)

**S12 Table. Best regions for predicting True Positives (TP, i.e. true diagnosis of Autism) for boys aged 5 to 10.**
(CSV)

**S13 Table. Best regions for predicting True Positives (TP, i.e. true diagnosis of Autism) for boys aged 10 to 15.**
(CSV)

**S14 Table. Best regions for predicting True Positives (TP, i.e. true diagnosis of Autism) for boys aged 15 to 20.**
(CSV)

**S15 Table. Best regions for predicting True Positives (TP, i.e. true diagnosis of Autism) for boys aged 20 to 64.**
(CSV)

**S16 Table. Best regions for predicting True Positives (TP, i.e. true diagnosis of Autism) for girls aged 5 to 10.**
(CSV)

**S17 Table. Best regions for predicting True Positives (TP, i.e. true diagnosis of Autism) for girls aged 10 to 15.**
(CSV)

**S18 Table. Best regions for predicting True Positives (TP, i.e. true diagnosis of Autism) for boys aged 15 to 20.**
(CSV)

**S19 Table. Best regions for predicting True Positives (TP, i.e. true diagnosis of Autism) for girls aged 20 to 64.**
(CSV)

**S20 Table. Comparing accuracy scores between data collection sites.**
(CSV)

**S1 Fig. Validation set accuracy during training for the two models DenseNet161 and Med3d-ResNet50.** DenseNet121 tended to have higher accuracy on the validation set than Med3d-ResNet50.
(PNG)

**S2 Fig. True and False Positives/Negatives for top three models: Med3DNet-ResNet50 (42 epochs), DenseNet121 (32 epochs), and DenseNet121 (70 epochs).**
(PNG)

**S3 Fig. Comparison of model predictions across datasets without comorbidity: Training, validation, and test sets.**
(PNG)

**S4 Fig. ROC AUC and accuracy scores by age group (5–10, 10–15, 15–20, 20–64) and gender (Male/Female) for models ResNet50 (42 epochs) and DenseNet121 (32 and 70 epochs) across datasets (Training, Validation, Testing, Testing 2).**
(PNG)

**S5 Fig. Comparison of social interaction Z-scores between False Negatives (FN), True Positives (TP), True Negatives (TN) and False Positives (FP).**
(PNG)

**S6 Fig. Probability scores by category for Med3d-ResNet50-42ep, based on SRS T-Scores in ABIDE II.**
(PNG)

**S7 Fig. Probability scores by category for DenseNet121-32ep, based on SRS T-Scores in ABIDE II.**
(PNG)

**S8 Fig. Probability scores by category for DenseNet121-70ep, based on SRS T-Scores in ABIDE II.**
(PNG)

**S9 Fig. Comparison of Med3d-ResNet50-42ep probability scores and SRS T-Score categories across different sites.**
(PNG)

**S10 Fig. Comparison of DenseNet121-32ep probability scores and SRS T-Score categories across various sites.**
(PNG)

**S11 Fig. Comparison of DenseNet121-70ep probability scores with SRS T-Score categories across various sites.**
(PNG)

**S12 Fig. Comparison of a slice (a) Before Preprocessing and (b) After Preprocessing.** Original voxel size was 1.2mm*1mm*1mm. Original image size was 160*240*256.
(PNG)

**S13 Fig. HighRes3DNet segmentation on OASIS 0145 brain scan.**
(PNG)

## Acknowledgments

We have much gratitude and appreciation for the Irish Research Council and the Hypercube Institute (Paris) for their funding.

We also thank Prof. Rhodri Cusack, Dr. Ariel Rokem, Dr. Robert Whelan and Prof. Louise Gallagher for their feedback and guidance.

## Author Contributions

**Conceptualization:** Mélanie Garcia.

**Data curation:** Mélanie Garcia.

**Formal analysis:** Mélanie Garcia.

**Funding acquisition:** Mélanie Garcia, Clare Kelly.

**Investigation:** Mélanie Garcia.

**Methodology:** Mélanie Garcia.

**Project administration:** Clare Kelly.

**Resources:** Mélanie Garcia, Clare Kelly.

**Software:** Mélanie Garcia.

**Supervision:** Clare Kelly.

**Validation:** Mélanie Garcia.

**Visualization:** Mélanie Garcia.

**Writing – original draft:** Mélanie Garcia, Clare Kelly.

**Writing – review & editing:** Clare Kelly.

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
