## [Decision Letter · Decision Letter 0]

2 Jul 2023

PONE-D-22-28394Towards 3D Deep Learning for neuropsychiatry: predicting Autism diagnosis using an interpretable Deep Learning pipeline applied to minimally processed structural MRI data

5946 Words, 2 Figures and 1 TablesPLOS ONE

Dear Dr. Garcia,

Thank you for submitting your manuscript to PLOS ONE. After careful consideration, we feel that it has merit but does not fully meet PLOS ONE’s publication criteria as it currently stands. Therefore, we invite you to submit a revised version of the manuscript that addresses the points raised during the review process.

We look forward to receiving your revised manuscript.

Kind regards,

Francesca Benuzzi, Ph.D.

Academic Editor

PLOS ONE

Journal Requirements:

2. Our internal editors have looked over your manuscript and determined that it is within the scope of our Reproducibility and Replicability in Neuroscience and Mental Health Research Call for Papers. The Collection will encompass a diverse and interdisciplinary set of protocols and research articles adhering to transparent and reproducible reporting practices in the areas of clinical psychology, psychiatry, mental health, and neuroscience. Additional information can be found on our announcement page: https://collections.plos.org/call-for-papers/reproducibility-and-replicability-in-neuroscience-and-mental-health-research/. If you would like your manuscript to be considered for this collection, please let us know in your cover letter and we will ensure that your paper is treated as if you were responding to this call. If you would prefer to remove your manuscript from collection consideration, please specify this in the cover letter.

Additional Editor Comments:

Dear authors,

I have sent your paper to three reviewers, and two of them have recommended major revisions but, I firmly believe that the issues raised can be addressed.

In particular, Reviewer 2 recommends greater transparency and accuracy in describing the methods, so that even those who are not familiar with the subject matter can fully understand the methodological aspects of the work.

Reviewer 3, on the other hand, emphasizes the need for a thorough revision of the text and figures.

Best regards,

Reviewers' comments:

Reviewer's Responses to Questions

**Comments to the Author**

1. Is the manuscript technically sound, and do the data support the conclusions?

Reviewer #1: No

Reviewer #2: Yes

Reviewer #3: Yes

2. Has the statistical analysis been performed appropriately and rigorously? 

Reviewer #1: No

Reviewer #2: I Don't Know

Reviewer #3: Yes

3. Have the authors made all data underlying the findings in their manuscript fully available?

Reviewer #1: Yes

Reviewer #2: Yes

Reviewer #3: Yes

4. Is the manuscript presented in an intelligible fashion and written in standard English?

Reviewer #1: No

Reviewer #2: Yes

Reviewer #3: Yes

5. Review Comments to the Author

Reviewer #1: The authors propose an AI method, based on 3D Deep learning algorithms, for the classification of ASD vs typical subjects. They used public datasets such as ABIDE I and ABIDE II together with ADHD200.

There are in my opinion some important issues:

-the authors explicitly avoid to perform spatial normalization to a common template. The reason for this choice is that additional post processing steps may introduce biases. However, a good spatial normalization is a powerful way to favor 3D algorithms convergence and to make importance features mapping interpretable.

Indeed, the authors process the gradCAM importance maps to be interpretable by adding an additional segmentation step and looking for correspondences between thresholded maps and segmentation which is quite complicated (and prone to biases such as those they tried to avoid).

In addition, images acquired in the same site may be acquired with a standard positioning protocol which will be very peculiar for 3D classifier such those trained by the authors. In the case of not well-balanced samples (for example with a predominance of ASD subjects in an acquisition site, this would imply a strong site biases which may be confused by an ASD effect.)

-the performances on the test sets are very low and almost comparable with the chance level. Since the test sets are the only data on which a classifier demonstrates its capability to generalize, the throughout discussion on the importance features does not reflect the real brain areas by which the classifier is guided to distinguish ASD vs TD but the areas by which the classifier managed to optimize its performance on the training and validation data.

-In S10 figure 8 it is clear that, despite the ASD gravity of deficit the performance of the classifier is completely dominated by the effect of site.

-The authors did not show any MR brain images before and after brain extraction and intensity normalization. I suggest to show for at least 3 examples taken from different acquisition sites the raw images, the “brain extracted” images and the intensity normalized images to better understand the real input of the 3D deep classifiers.

-The authors did not show any example of gradCAM maps and brain segmentation. This is important to understand how much focused are the maps of importance in comparison with the ROI segmentations.

-There are some parts in the results section for which the correspondent description in the methods section is missing (the effect of gender, the effect of age, the multi-site effect…).

-S1 table 1 and 2. Please reduce the significant figures.

- S10 Fig 7, S10 Fig 8, and S10 Fig 9 please limit the x axis to 1.

Reviewer #2: This is a nicely written paper but the emphasis does not seem quite right. I am not a computational scientist but I do understand this technique in a broad way in line with PLOS 1's wide scope. So some issues may reflect my lack of knowledge on what is standard in the literature on machine learning.

1, I understand this approach takes a lot of computational resources but computer time used is not the same burden as human time used. So I don't buy that saving processing time by not morphing the brains into MNI space is a major +. It makes more sense to do it for substantive reasons (i.e, capture anomalies about brain structure in ASD - but what? gyrification in temporal lobe?).

2. One take home I got was - these machine learning approaches on structural data including deep learning do not do a great job predicting autism. Site seems to matter - so maybe that is a real thing as culture does seem to affect behavioral differences in autism (koh & Milne).

3. A major plus is going back to identify the relevant structures. However,

a. you do not give many references or ideas why these structures are important except for language, which is only true of some of them (for instance, TPJ = mentalizing), or whether/how they differ in ASD (based on references).

b. you do not tell us how these structures differ in your data, but it seems that you cannot, right? COuld you test one theoretical prediction to examine this? If you have an idea of how they might differ (ie thicker frontal cortex).

4. You present too many options for models, how to identify relevant areas etc. Decide which you think is (most) correct, and minimize discussion of the other options. Can you identify if the differences in specificity and sensitivity between models are significantly different and identify the best one? This also makes the figures too busy.

5. A big question - why would you want to use expensive, hard to collect neuroimaging data to identify autism? you are likely to get better, cheaper predictive models from eye movements to a set of pictures or other behavioral measures. So I am not convinced we would ever use fMRI to identify autism, but if you can go back and identify the relevant regions and how they differ - that does seem useful.

6. Another major plus are the age and gender analyses. Perhaps expand a bit further, illustrate age, and clarify if they interact at all. why do predictive regions change with age? Are there typical changes in these age ranges that differ in ASD? of course more examination of the effect of IQ and comorbid conditions (like ADHD) would also be great.

7. I found these items jargony/ did not understand the concept fully:

a. Guided Grad-CAM

b. not sure why you would ever use a 2-D program for 3-D brain data?

c. would like more details in why gyrification differs in autism and how, since this is a good reason to not morph brain data (i.e., can change side of gyrus activation is evident on).

d. differences/similarities between Machine learning vs Deep learning vs traditional multivariate approaches

e. not sure what cropping and padding is - just adding on to the edges?

f. I liked the discussion of false positives, negatives, etc but it gets a bit confusing as to the actual implications? Perhaps you could clarify with a table or just clarify writing?

g. section 4.4 does not make sense to me, in light of already using BET software (removing info outside the brain).

f. not sure what the inference step is?

Nice job in transparency and using large datasets.

Reviewer #3: In summary, this manuscript aims to address the strong data heterogeneity and interpretability of deep learning in predicting Autism diagnosis based on structured MRI data and then proposes an interpretable predictive pipeline for inferring Autism diagnosis using 3D Deep Learning applied to minimally processed structural MRI scans. The design of the solution looks like reasonable and the analyses on experiments results are justified. However, the manuscript must be significantly revised and address the following issues.

1. The manuscript needs to be carefully revised from beginning to end. There are language errors and it needs to be polished by professional language experts.

2. The authors claim that one contribution of the study is the adoption of the minimal preprocessing pipeline , i.e. no transformation to template space, to avoid any impact of brain normalization on the detecting of Autism-related alterations in brain structure. However, the manuscript lacks a comparison with the relevant methods of using spatial template matching, which is not rigorous enough.

3. The resolution of the graphics in the current manuscript is too low, please replace it with high-quality graphics.

4. A large number of figures, tables, and experimental results are attached to the supporting material, which are important for method demonstration and should be included in the body, such as Table S5, S6, etc.

5. The proposed solution to the problem of data heterogeneity in the manuscript was achieved by training large data sets, which, in my opinion, did not completely or even alleviate the relevant problems, and therefore the description of this contribution should be removed. At the same time, to alleviate the problem of data heterogeneity, the author can explore the direction of reducing the distribution difference between the source domain and the target domain.

6. The performance of the autism diagnostic task in the manuscript was modest and lacked comparison with existing and conventional methods.

6. PLOS authors have the option to publish the peer review history of their article (what does this mean?). If published, this will include your full peer review and any attached files.

Reviewer #1: No

Reviewer #2: No

Reviewer #3: No

---

## [Author Response · Author response to Decision Letter 0]

8 Jan 2024

PONE-D-22-28394

Towards 3D Deep Learning for neuropsychiatry: predicting Autism diagnosis using an interpretable Deep Learning pipeline applied to minimally processed structural MRI data

5946 Words, 2 Figures and 1 Tables

PLOS ONE

Dear Dr. Garcia,

Thank you for submitting your manuscript to PLOS ONE. After careful consideration, we feel that it has merit but does not fully meet PLOS ONE’s publication criteria as it currently stands. Therefore, we invite you to submit a revised version of the manuscript that addresses the points raised during the review process.

● A rebuttal letter that responds to each point raised by the academic editor and reviewer(s). You should upload this letter as a separate file labeled 'Response to Reviewers'.

● A marked-up copy of your manuscript that highlights changes made to the original version. You should upload this as a separate file labeled 'Revised Manuscript with Track Changes'.

● An unmarked version of your revised paper without tracked changes. You should upload this as a separate file labeled 'Manuscript'.

 We look forward to receiving your revised manuscript.

Kind regards,

Francesca Benuzzi, Ph.D.

 Academic Editor

 PLOS ONE

Journal Requirements:

2. Our internal editors have looked over your manuscript and determined that it is within the scope of our Reproducibility and Replicability in Neuroscience and Mental Health Research Call for Papers. The Collection will encompass a diverse and interdisciplinary set of protocols and research articles adhering to transparent and reproducible reporting practices in the areas of clinical psychology, psychiatry, mental health, and neuroscience. Additional information can be found on our announcement page: https://collections.plos.org/call-for-papers/reproducibility-and-replicability-in-neuroscience-and-mental-health-research/. If you would like your manuscript to be considered for this collection, please let us know in your cover letter and we will ensure that your paper is treated as if you were responding to this call. If you would prefer to remove your manuscript from collection consideration, please specify this in the cover letter.

Yes, we would like our manuscript to be considered for this collection.

We have included a new section 2.6 Ethics statement (p15).

 Additional Editor Comments:

 Dear authors,

 I have sent your paper to three reviewers, and two of them have recommended major revisions but, I firmly believe that the issues raised can be addressed.

 In particular, Reviewer 2 recommends greater transparency and accuracy in describing the methods, so that even those who are not familiar with the subject matter can fully understand the methodological aspects of the work.

 Reviewer 3, on the other hand, emphasizes the need for a thorough revision of the text and figures.

 Best regards,

 Reviewers' comments:

 Reviewer's Responses to Questions

Comments to the Author

 1. Is the manuscript technically sound, and do the data support the conclusions?

Reviewer #1: No

Reviewer #2: Yes

Reviewer #3: Yes

2. Has the statistical analysis been performed appropriately and rigorously?

Reviewer #1: No

Reviewer #2: I Don't Know

Reviewer #3: Yes

3. Have the authors made all data underlying the findings in their manuscript fully available?

Reviewer #1: Yes

Reviewer #2: Yes

Reviewer #3: Yes

4. Is the manuscript presented in an intelligible fashion and written in standard English?

Reviewer #1: No

Reviewer #2: Yes

Reviewer #3: Yes

5. Review Comments to the Author

Reviewer #1: The authors propose an AI method, based on 3D Deep learning algorithms, for the classification of ASD vs typical subjects. They used public datasets such as ABIDE I and ABIDE II together with ADHD200.

 There are in my opinion some important issues:

 -the authors explicitly avoid to perform spatial normalization to a common template. The reason for this choice is that additional post processing steps may introduce biases. 

However, a good spatial normalization is a powerful way to favor 3D algorithms convergence and to make importance features mapping interpretable.

Thank you for these comments on our approach. The goal of this paper is indeed to propose and provide an initial test of an alternative to using template registration when working with neurodivergent populations, where the risk of distorting data and introducing significant biases is higher. We agree that 3D algorithms may converge faster when all brains have the same shape and size. However, in our view, there is limited utility in having an algorithm converge well if the trade-off is the potential for distortion and biased metrics. In this work, we demonstrated the feasibility of an alternative pipeline that avoids this traditional preprocessing step. We do not claim that the proposed method is bias-free - in fact we emphasized the need to use multiple analytical algorithms, since each has its own biases. 

In section 4.3. (p30), we added a discussion about the fact that classification was done in native space rather than template space, along with a proposal on how the cost or benefit of native space analysis could be measured in section 4.4. (p32). 

Indeed, the authors process the gradCAM importance maps to be interpretable by adding an additional segmentation step and looking for correspondences between thresholded maps and segmentation which is quite complicated (and prone to biases such as those they tried to avoid).

We acknowledge that our interpretation approach may seem somewhat complicated, which is due to our goal of avoiding the spatial normalization step. A key step in our interpretation approach is the segmentation algorithm HighRes3DNet, a fast and readily implementable segmentation method available that avoids template registration. We combined this segmentation approach with guided GradCAM to enable interpretability. A similar interpretability analysis could be conducted on normalized brains, but for the reasons outlined above and in the paper, we opted to examine the feasibility of a pipeline that avoids normalization. Currently, there are limited alternatives to template registration for large-scale neuroimaging analyses. Our proof-of-concept work opens up possibilities for future research into developing and evaluating alternative pipelines that minimize potential sources of bias or distortion in research with neurodivergent populations. 

We mentioned this point in our new discussion paragraph in section 4.4. (p32).

In addition, images acquired in the same site may be acquired with a standard positioning protocol which will be very peculiar for 3D classifier such those trained by the authors. 

We are unsure about what the reviewer meant by this comment. To be clear about our methodology - our preprocessing of the raw 3D images was minimal, but did include an initial step in which each scan was reoriented to RAS+ orientation. This standardization of orientation was necessary so that all images used in training were consistent in terms of axes and orientation.

In the case of not well-balanced samples (for example with a predominance of ASD subjects in an acquisition site, this would imply a strong site biases which may be confused by an ASD effect.)

Thank you for raising this point. Indeed, we observed a multi-site effect, which could be attributed to several factors: differences in image acquisition protocols between sites (with the open question of which parameters have the biggest influence), varying levels of ASD severity across sites given the spectrum nature of ASD, differential sample size across sites (e.g., the algorithms were trained on 103 scans from NYU vs 5 from Stanford), unbalanced diagnosis/no-diagnosis representation within some sites (e.g. 27 ASD vs 76 non-ASD from NYU in the training set), comorbidities, gender and age distributions, and individual variability. Other potential contributors like geometrical factors related to the image data may also play a role.

We did not attempt to quantify the influence of these potential factors, since uneven sample sizes across sites posed a significant challenge and the meta-data available was insufficient. While we could have matched sample sizes across sites, this would have dramatically reduced the data set. We have added further discussion of these issues in section 4.3. (p30), acknowledging in section 4.4. (p32) that further investigations with larger, balanced datasets are needed to fully characterize the source(s) of site effects.

 -the performances on the test sets are very low and almost comparable with the chance level. Since the test sets are the only data on which a classifier demonstrates its capability to generalize, the throughout discussion on the importance features does not reflect the real brain areas by which the classifier is guided to distinguish ASD vs TD but the areas by which the classifier managed to optimize its performance on the training and validation data.

Thank you for your detailed feedback and the observations regarding the classifier's performance on the test sets. We agree that test sets are an important assessment of a model's capability to generalize. In our study, although test set performance could certainly be improved, it was consistent with that achieved by previous studies using Machine Learning models trained on sMRI data. It is important to note that we do not view this work as a final product but rather as an ongoing project that requires continuous improvement. As noted above, one goal was to establish the feasibility of a prediction pipeline that omitted spatial normalization. Our models’ comparable performance suggests that this is an analytic approach that is worth refining. We have openly shared all our code to foster collaboration, improvement, and reuse in other neuroimaging applications. We sincerely hope the community can build upon or adapt our work for refined applications.

We are unsure about what was meant by “the importance features does not reflect the real brain areas by which the classifier is guided to distinguish ASD vs TD but the areas by which the classifier managed to optimize its performance on the training and validation data”. We chose the guided grad-CAM algorithm (over grad-CAM) for its capability to provide class-specific output maps of voxel importance. We believe this is a useful way to identify and describe the anatomical brain patterns that contribute to accurate predictions of autism diagnosis (i.e., True Positives). Improving our model’s performance may result in changes to the set of regions identified, because it may open the model’s understanding to a wider range of data information, potentially comprising a greater variety of scanning protocols, better representativeness of the autism spectrum, etc. 

 -In S10 figure 8 it is clear that, despite the ASD gravity of deficit the performance of the classifier is completely dominated by the effect of site.

We agree that there is a clear site effect; we provided further evidence and illustration of this in S31 Table. However, it is important to note that S28, S29, S30 Figs represent just one diagnostic scale for ASD - SRS scores for social responsiveness. ASD diagnosis incorporates many other factors, such as those related to verbal/non-verbal communication scales, repetitive behavior metrics, etc. Examining algorithm performance across sites using these additional diagnostic dimensions may reveal better discrimination than the SRS scale alone and improve generalizability. 

The multi-site prediction discrepancy highlights the ongoing challenge posed by site effects, which likely reflect multiple contributing factors, as discussed above. Further investigation is needed to elucidate the influence of heterogeneous scanning protocols, ASD subtypes/severity, age distributions, comorbidities, and other factors underlying these site effects. This represents an important future direction for developing more robust, widely appli

---

## [Decision Letter · Decision Letter 1]

10 May 2024

PONE-D-22-28394R13D CNN for neuropsychiatry: predicting Autism with interpretable Deep Learning applied to minimally preprocessed structural MRI dataPLOS ONE

Dear Dr. Garcia,

Thank you for submitting your manuscript to PLOS ONE. After careful consideration, we feel that it has merit but does not fully meet PLOS ONE’s publication criteria as it currently stands. Therefore, we invite you to submit a revised version of the manuscript that addresses the points raised during the review process.

We look forward to receiving your revised manuscript.

Kind regards,

Yangsong Zhang, Ph.D.

Academic Editor

PLOS ONE

Journal Requirements:

Additional Editor Comments:

The authors should address the comments from the reviewers.

Reviewers' comments:

Reviewer's Responses to Questions

**Comments to the Author**

1. If the authors have adequately addressed your comments raised in a previous round of review and you feel that this manuscript is now acceptable for publication, you may indicate that here to bypass the “Comments to the Author” section, enter your conflict of interest statement in the “Confidential to Editor” section, and submit your "Accept" recommendation.

Reviewer #2: All comments have been addressed

Reviewer #3: All comments have been addressed

2. Is the manuscript technically sound, and do the data support the conclusions?

Reviewer #2: Partly

Reviewer #3: Yes

3. Has the statistical analysis been performed appropriately and rigorously? 

Reviewer #2: I Don't Know

Reviewer #3: Yes

4. Have the authors made all data underlying the findings in their manuscript fully available?

Reviewer #2: Yes

Reviewer #3: Yes

5. Is the manuscript presented in an intelligible fashion and written in standard English?

Reviewer #2: Yes

Reviewer #3: Yes

6. Review Comments to the Author

Reviewer #2: the brain bases of asd is a messy confusing topic, although there are some consistencies in the literature. This paper is a good attempt but I am not sure that the authors have convinced us that looking for a single, brain signature makes sense for diagnosing a behaviorally-defined and heterogeneous disorder with many genetic bases

Reviewer #3: All concerns raised previously have been properly addressed. The paper is now in a better shape and ready for acceptance.

7. PLOS authors have the option to publish the peer review history of their article (what does this mean?). If published, this will include your full peer review and any attached files.

Reviewer #2: No

Reviewer #3: No

---

## [Author Response · Author response to Decision Letter 1]

23 Jun 2024

PONE-D-22-28394R1

3D CNN for neuropsychiatry: predicting Autism with interpretable Deep Learning applied to minimally preprocessed structural MRI data

PLOS ONE

Dear Dr. Garcia,

Thank you for submitting your manuscript to PLOS ONE. After careful consideration, we feel that it has merit but does not fully meet PLOS ONE’s publication criteria as it currently stands. Therefore, we invite you to submit a revised version of the manuscript that addresses the points raised during the review process.

● A rebuttal letter that responds to each point raised by the academic editor and reviewer(s). You should upload this letter as a separate file labeled 'Response to Reviewers'.

● A marked-up copy of your manuscript that highlights changes made to the original version. You should upload this as a separate file labeled 'Revised Manuscript with Track Changes'.

● An unmarked version of your revised paper without tracked changes. You should upload this as a separate file labeled 'Manuscript'.

We look forward to receiving your revised manuscript.

Kind regards,

Yangsong Zhang, Ph.D.

Academic Editor

PLOS ONE

Journal Requirements:

Additional Editor Comments:

The authors should address the comments from the reviewers.

Reviewers' comments:

Reviewer's Responses to Questions

Comments to the Author

1. If the authors have adequately addressed your comments raised in a previous round of review and you feel that this manuscript is now acceptable for publication, you may indicate that here to bypass the “Comments to the Author” section, enter your conflict of interest statement in the “Confidential to Editor” section, and submit your "Accept" recommendation.

Reviewer #2: All comments have been addressed

Reviewer #3: All comments have been addressed

2. Is the manuscript technically sound, and do the data support the conclusions?

Reviewer #2: Partly

Reviewer #3: Yes

3. Has the statistical analysis been performed appropriately and rigorously?

Reviewer #2: I Don't Know

Reviewer #3: Yes

4. Have the authors made all data underlying the findings in their manuscript fully available?

Reviewer #2: Yes

Reviewer #3: Yes

5. Is the manuscript presented in an intelligible fashion and written in standard English?

Reviewer #2: Yes

Reviewer #3: Yes

6. Review Comments to the Author

Reviewer #2: the brain bases of asd is a messy confusing topic, although there are some consistencies in the literature. This paper is a good attempt but I am not sure that the authors have convinced us that looking for a single, brain signature makes sense for diagnosing a behaviorally-defined and heterogeneous disorder with many genetic bases

Thank you for your feedback. We appreciate your acknowledgment of the complexities surrounding the brain bases of ASD and the challenges inherent in this research area. We understand the concern regarding the search for a single brain signature for a behaviorally-defined and heterogeneous disorder like ASD, which indeed has many genetic bases. We extensively talked about this issue in the Introduction and Discussion.

In our study, our aim was to contribute to the ongoing effort to identify consistent neural patterns associated with ASD, while acknowledging the inherent variability within the disorder. We agree that expecting a single, definitive brain signature may be overly simplistic given the heterogeneity of ASD. Instead, our approach is to identify convergent patterns that could aid in the development of more nuanced diagnostic tools. We now mentioned this last point in the Introduction (page 6).

Reviewer #3: All concerns raised previously have been properly addressed. The paper is now in a better shape and ready for acceptance.

7. PLOS authors have the option to publish the peer review history of their article (what does this mean?). If published, this will include your full peer review and any attached files.

Do you want your identity to be public for this peer review? For information about this choice, including consent withdrawal, please see our Privacy Policy.

Reviewer #2: No

Reviewer #3: No

---

## [Decision Letter · Decision Letter 2]

8 Aug 2024

3D CNN for neuropsychiatry: predicting Autism with interpretable Deep Learning applied to minimally preprocessed structural MRI data

PONE-D-22-28394R2

Dear Dr. Garcia,

We’re pleased to inform you that your manuscript has been judged scientifically suitable for publication and will be formally accepted for publication once it meets all outstanding technical requirements.

Kind regards,

Yangsong Zhang, Ph.D.

Academic Editor

PLOS ONE

Additional Editor Comments (optional):

The manuscript was significantly improved after two round revision, and can be accepted for publication.

Reviewers' comments:

Reviewer's Responses to Questions

**Comments to the Author**

1. If the authors have adequately addressed your comments raised in a previous round of review and you feel that this manuscript is now acceptable for publication, you may indicate that here to bypass the “Comments to the Author” section, enter your conflict of interest statement in the “Confidential to Editor” section, and submit your "Accept" recommendation.

Reviewer #2: All comments have been addressed

2. Is the manuscript technically sound, and do the data support the conclusions?

Reviewer #2: Yes

3. Has the statistical analysis been performed appropriately and rigorously? 

Reviewer #2: I Don't Know

4. Have the authors made all data underlying the findings in their manuscript fully available?

Reviewer #2: Yes

5. Is the manuscript presented in an intelligible fashion and written in standard English?

Reviewer #2: Yes

6. Review Comments to the Author

Reviewer #2: Thanks for addressing my concerns. I still wonder how to address the diversity in autism but it is a difficult topic.

7. PLOS authors have the option to publish the peer review history of their article (what does this mean?). If published, this will include your full peer review and any attached files.

Reviewer #2: No

---

## [Editor Report · Acceptance letter]

21 Aug 2024

PONE-D-22-28394R2 

PLOS ONE

Dear Dr. Garcia, 

I'm pleased to inform you that your manuscript has been deemed suitable for publication in PLOS ONE. Congratulations! Your manuscript is now being handed over to our production team.

Kind regards, 

on behalf of

Prof. Yangsong Zhang 

Academic Editor

PLOS ONE